# Pharmacotherapy Problems in Best Possible Medication History of Hospital Admission in the Elderly

**DOI:** 10.3390/pharmacy10050136

**Published:** 2022-10-18

**Authors:** Ivana Marinović, Ivana Samardžić, Slaven Falamić, Vesna Bačić Vrca

**Affiliations:** 1Department of Clinical Pharmacy, Clinical Hospital Dubrava, 10000 Zagreb, Croatia; 2Pharmacy Popović, 31551 Belišće, Croatia; 3Faculty of Pharmacy and Biochemistry, 10000 Zagreb, Croatia

**Keywords:** transfer of care, elderly patients, polypharmacy, potentially inappropriate medications, drug-drug interactions, renal risk drugs, clinical pharmacist

## Abstract

Transfer of care is a sensitive process, especially for the elderly. Polypharmacy, potentially inappropriate medications (PIMs), drug-drug interactions (DDIs), and renal risk drugs (RRDs) are important issues in the elderly. The aim of the study was to expand the use of the Best Possible Medication History (BPMH) and to evaluate polypharmacy, PIMs, DDIs, and inappropriately prescribed RRDs on hospital admission, as well as to determine their mutual relationship and association with patients’ characteristics. An observational prospective study was conducted at the Internal Medicine Clinic of Clinical Hospital Dubrava. The study included 383 elderly patients. Overall, 49.9% of patients used 5–9 prescription medications and 31.8% used 10 or more medications. EU(7)-PIMs occurred in 80.7% (n = 309) of the participants. In total, 90.6% of participants had ≥1 potential DDI. In total, 43.6% of patients were found to have estimated glomerular filtration rate < 60 mL/min/1.73 m^2^, of which 64.7% of patients had one or more inappropriately prescribed RRDs. The clinical pharmacist detected a high incidence of polypharmacy, PIMs, DDIs, and inappropriately prescribed RRDs on hospital admission. This study highlights the importance of early detection of pharmacotherapy problems by using the BPMH in order to prevent their circulation during a hospital stay. The positive correlations between polypharmacy, PIMs, DDIs, and inappropriately prescribed RRDs indicate that they are not independent, but rather occur simultaneously.

## 1. Introduction

Transfer of care is a vulnerable process that increases the risk of drug-related problems, especially in the elderly. The World Health Organisation (WHO) launched the third Global Patient Safety Challenge with aim of improving the medication safety, “Medication Without Harm”, where the key areas of the challenge represent the transfer of care, polypharmacy, and high-risk situations. High-risk situations include hospital treatment, and drug management in elderly patients, as well as those with concomitant renal impairment. The WHO’s goal is to reduce severe avoidable harm related to medications by 50% [1].

More than 20% of the European Union (EU) population are the elderly, and by 2050, this figure will double at the global level [2,3]. Population aging, higher incidence of chronic diseases, and the application of multiple guidelines are some of the reasons for the growing trend of prescribing drugs. Polypharmacy, usually defined as the use of ≥5 medications, is rising, especially in the elderly [4]. Polypharmacy is the main risk factor for adverse drug events (ADE)-related hospitalization in older adults and has been linked to the high risk of potentially inappropriate medication (PIMs) use and drug-drug interactions (DDIs) [5,6]. PIMs are defined as medications whose adverse risks outreach their positive therapeutic effects when compared to alternative therapies [7]. The use of PIMs is prevalent among the elderly and is affiliated with an increased risk of adverse health outcomes [8]. Apart from PIMs, DDIs represent a major problem for therapy management, especially in the elderly, as they can compromise patient safety [6]. It is estimated that DDIs cause up to 5% of hospitalizations in elderly patients [9].

Treatment of chronic kidney disease (CKD) represents a global health burden with a significant share of health care costs [10]. Older adults with CKD are highly vulnerable to polypharmacy [11]. The aging process and renal impairment modify the clinical drug profile, which results in a 3–10 times higher incidence of adverse drug reactions in older adults with CKD than in those without it [12]. In order to avoid the occurrence of ADEs, optimal drug selection and dosing modification should be carried out for renally cleared and potentially nephrotoxic drugs in the elderly. The recent systematic review reported the use of a contraindicated medication or inappropriately high dose according to kidney function ranging from 9.4% to 81.1% [13].

With the length of hospital stay the number of medications increases, which complicates pharmacotherapy management and jeopardizes patients’ safety [14]. The detection of pharmacotherapy problems upon admission to the hospital is important for further pharmacotherapy optimization, especially when new medications are introduced into therapy.

Medication reconciliation is a relevant safety procedure in medication management during the transition of care which has also been proven as a cost-saving measure [15]. Medication reconciliation is the process of creating the Best Possible Medication History (BPMH) and comparing the list with orders written at each transition of care. BPMH is an accurate and thorough list of medications that a patient is taking, including dose, frequency, and administration route, which may vary from other healthcare records [16]. Guidance states that BPMH in an acute setting should be completed as soon as possible, within 24 hours, upon transfer of care [17]. BPMH is a useful instrument for detecting unintentional pharmacotherapy discrepancies (drug omission, addition, substitution, incorrect dose, frequency, and route of administration) [18,19]. However, the use of BPMH should be expanded for the detection of a wider spectrum of the above-mentioned pharmacotherapy problems (polypharmacy, PIMs, DDIs, renal risk drugs-RRDs) in order to increase patient safety.

The aim of the study was to evaluate polypharmacy, PIMs, DDIs, and RRDs by using the BPMH on hospital admission and to determine their mutual relationship and association with patient characteristics.

## 2. Materials and Methods

### 2.1. Setting and Participants

An observational prospective study was conducted at the Internal Medicine Clinic of Clinical Hospital Dubrava. Clinical Hospital Dubrava is a tertiary care, 600-bed teaching institution whose emergency service provides care to a population of approximately 350,000 inhabitants. The Internal Medicine Clinic consists of eight departments: Department for cardiovascular medicine, Department of endocrinology, diabetes, diseases of metabolism and clinical pharmacology, Department of gastroenterology, hepatology and clinical nutrition, Department of Rheumatology and Clinical Immunology, Department of nephrology and dialysis, Department of pulmonology, Department of haematology and Department of intensive care medicine. Patients aged 65 or older were eligible to participate if they had been admitted to the Internal Medicine Clinic in the period from December 2018 to March 2020 and were willing to sign the informed consent (personally or through their caregiver). Participants were randomly selected, using a computer-generated random number table. The sample size was based on data availability in this period.

### 2.2. Data Collection

The hospital clinical pharmacist created the BPMH for each patient within 24 h of admission. The standardized process of obtaining BPMH was in accordance with the Protocol on Medication Reconciliation and Implementation Guide [20]. By patient interview, a clinical pharmacist obtained a thorough pharmacotherapy history. BPMH includes the drug name, dosage, frequency, and route of administration [16]. In addition to prescription medications, BPMH also includes over-the-counter (OTC) drugs, vitamins, herbal preparations, dietary supplements, and vaccines. Pharmacotherapy history was completed with information about relevant demographic and clinical data. Patients’ diagnoses were classified according to the International Classification of Diseases (ICD-10). All available sources of information used to obtain the BPMH were recorded, for example, previous medical records, discharge summaries, laboratory data, examination of medication vials, review of a home medication list, interview with caregiver, information obtained from the community pharmacist and primary care physician.

### 2.3. Outcome Measures

Patients’ therapies in BPMH were listed according to the Anatomical Therapeutic Chemical (ATC) Classification System. The use of prescribed medications in BPMH was divided into the following categories: low medication use (<5 medications), polypharmacy (5–9 medications), and excessive polypharmacy (≥10 medications) [21]. The EU(7)-PIM list was used to detect PIMs in BPMHs. The EU(7)-PIM list is a European list for PIM based on different national PIM lists (German PRISCUS list, PIM lists from USA, France, and Canada) published in 2015. The EU(7)-PIM list comprises 282 chemical substances from 34 therapeutic groups [7]. 

DDI analysis included prescribed medications and OTC drugs in BPMH. The Lexicomp® Lexi-InteractTM Online (Lexi-Comp, Inc., Hudson, USA) screening program was used for pharmacotherapy analysis. Lexicomp classifies potential DDIs according to clinical significance in 5 classes (A, B, C, D, X). Classes C, D and X, interpreted as “monitor drug therapy”, “consider therapy modification” and “avoid combination” are clinically significant and are included in the analysis.

Data for renal function was documented upon hospital admission. Kidney Disease: Improving Global Outcomes (KDIGO) classification of renal impairment includes following categories G1–G5 based on the estimated glomerular filtration rate (eGFR) values: stage G1 (eGFR ≥ 90 mL/min/1.73 m^2^), stage G2 (eGFR 60–89 mL/min/1.73 m^2^), G3a (eGFR 45–59 mL/min/1.73 m^2^), G3b (eGFR 30–44 mL/min/1.73 m^2^), G4 (eGFR 15–29 mL/min/1.73 m^2^), and G5 (eGFR < 15 mL/min/1.73 m^2^) [22]. The eGFR was calculated by using the CKD-epidemiology collaboration (CKD-EPI) formula. Patients pertaining to stages 3a, 3b, 4, and 5 were considered to have renal impairment. Drugs that required dose adjustment and drugs with contraindicated use with regard to renal impairment were designated as RRDs. RRDs with unadjusted doses and contraindications were marked as inappropriately prescribed RRDs. Dose adjustment and contraindication according to renal function were determined using the Summary of Product Characteristics. In the RRDs analysis, fixed-dose drug combinations were observed as one drug with an accompanying ATC code. 

### 2.4. Statistical Analysis

R Core Team (Software R 4.2.0, Vienna, Austria, 2022) was used for data management and analyses [23]. Standard descriptive statistics were used to describe demographic and clinical data of the study population, number, and types of determined PIMs, DDIs, and RRDs. Frequencies and percentages were used to describe categorical data while the median and interquartile range (IQR) were used to present continuous variables. To analyze the relationship between the criterion variables, the Spearman correlation coefficient was calculated using the “rcorr” function within the Hmisc package for R Core Team (Software R 4.2.0, Vienna, Austria, 2022) [24]. It should be noted that due to the collinearity within polypharmacy, this variable was coded into three levels–No/Polypharmacy/Excessive polypharmacy. To examine the combined effects of predictor variables on the selected criteria, a logistic regression was used. The analyses were conducted with the glm function within R Core Team (Software R 4.2.0, Vienna, Austria, 2022). The Nagelkerke and Cox and Snell’s pseudo R 2 were calculated using DescTools package [25]. The level of significance for statistical tests, the *p*-value, was set at *p* < 0.05.

## 3. Results

### 3.1. Patients Characteristics

The research included 383 elderly patients with a median age of 76 (70–80). A total of 43.6% of patients were found to have eGFR < 60 mL/min/1.73 m^2^ (KDIGO stages 3a, 3b, 4 and 5). Detailed patients characteristics are shown in Table 1.

### 3.2. Polypharmacy

Overall, 49.9% of patients used 5–9 prescription medications, and 31.8% of patients used 10 or more medications. The median number of medications in the BPMH per patient was 8 (5–11). The number of OTC drugs and dietary supplements reported in patients’ BPMH was 98 and 88, respectively. 

The patterns of medication use and frequency were presented in Table 1. The most frequent ATC drug classes in BPMH were groups C cardiovascular system (C), alimentary tract and metabolism(A), nervous system (N), and blood and blood-forming organs(B). Drugs with antihypertensive effects were the most commonly used medications, with angiotensin-converting enzyme inhibitors (ACEI) as the most common antihypertensive class (n = 195). 

### 3.3. PIMs

PIMs based on EU(7)-PIM criteria occurred in 80.7% (n = 309) of the patients. A total of 689 PIMs were reported in BPMHs with 56 different drugs. The average number of PIMs per patient was 1.8. The most common PIMs were pantoprazole, diazepam, tramadol and moxonidine (Table 2).

### 3.4. DDIs

Overall, 2629 potentially clinically significant drug interactions were determined; 2270 (86.3%) of them required increased patient monitoring, while 323 (12.3%) interactions required specific therapy modification and 36 (1.4%) should be avoided. In total, 90.6 % of patients had ≥1 potentially clinically significant DDI in their BPMH; 87.5% had at least one C interaction, 46.7% had at least one D interaction and 8.1% of patients had at least one X interaction. The overall rate of clinically significant DDIs was 6.9 per patient. The most common potential clinically significant DDIs are shown in Table 3. Drugs with antihypertensive effects were the most commonly involved in determined clinically significant interactions.

### 3.5. RRDs

A total of 279 RRDs were reported in BPMH in patients with renal impairment. There were 183 drugs needing dose adjustment and 96 drugs with a contraindication with regard to renal function. For 47% (n = 86) of drugs needing dose adjustment, the dose was adjusted in BPMH with regards to patients’ renal function, while more than half of drugs (53%, n = 97) did not have an adjusted dose. Of all patients with renal impairment, 64.7% had one or more inappropriately prescribed RRDs in their BPMH (unadjusted dose and contraindicated drugs); 32.3% of patients had one or more contraindicated drugs with regards to their renal function; 47.9% of patients had an unadjusted drug dose and 41.3% of patients had an adjusted drug dose. The most represented RRDs were: perindopril (separately and in fixed combinations), moxonidine, metformin, and ramipril (Table 4).

### 3.6. The Relationship between Polypharmacy, PIMs, DDIs, and Inappropriately Prescribed RRDs

The relationship between polypharmacy, PIMs, DDIs, and inappropriately prescribed RRDs is described in Table 5. Polypharmacy is positively correlated with PIMs, DDIs, and inappropriately prescribed RRDs. PIMs were positively correlated with DDIs, and DDIs were positively correlated with inappropriately prescribed RRDs. 

### 3.7. Factors Associated with Polypharmacy, PIMs, DDIs, and Inappropriately Prescribed RRDs

The association between patients’ characteristics and polypharmacy, PIMs, DDIs, and inappropriately prescribed RRDs is presented in Table 6. Patients with a more severe level of renal impairment, a higher number of diagnoses, and recent hospitalization were at higher risk for excessive polypharmacy. The number of prescription medications is a significant risk factor for PIMs, DDIs and inappropriately prescribed RRDs. Patients with recent hospitalization were at higher risk for PIM use. Females had a higher risk for D interactions and a lower risk for X interactions than men. Patients with a more severe level of renal impairment had a lower risk for X interactions. Female patients and patients with a more severe level of renal impairment were at higher risk for inappropriately prescribed RRDs. It should be noted that the residents in a nursing home had an extremely high and insignificant odds ratio, which is due to the low number of participants who have no variability in the criterion variable.

## 4. Discussion

Transfer of care is a sensitive process that increases the risk of ADEs, especially in the elderly. In our study, a clinical pharmacist, using multiple sources of information, obtained and evaluated BPMH for 383 hospitalized older patients admitted to the Internal Medicine Clinic. This research determined a very high incidence of polypharmacy, PIMs, DDIs, and inappropriately prescribed RRDs detected in BPMH. To the best of our knowledge, this is the first study to explore the incidence and types of polypharmacy, PIMs, DDIs, and inappropriately prescribed RRDs by using BPMH, and to determine their mutual relationship and association with patient characteristics.

A high level of polypharmacy was identified in in this research. According to a recent review by Khezrian et al., the prevalence of polypharmacy varied between 10% and 90% [26]. Polypharmacy represents one of the major challenges for the healthcare system. It has increased markedly in recent years and is still increasing as more people suffer from chronic diseases [26,27]. In this study, a higher number of medications was determined in the BPMH upon admission compared to the study conducted in 2016 in the same clinical setting but included the general population (8 vs. 6 per patient) [18]. Our regression analysis showed the association of excessive polypharmacy with the number of diagnoses in addition to impaired renal function and recent hospitalization. Patients with renal impairment often experience polypharmacy especially in the later stages of CKD when patients develop numerous metabolic complications that require the prescription of multiple drugs according to guidelines [11,28]. The association between recent hospitalizations and polypharmacy in elderly patients can be explained by the fact that patients with a weaker health status have more complex pharmacotherapy and experience hospitalizations more frequently [29]. In our research, polypharmacy positively correlated with detected pharmacotherapy problems: PIMs, DDIs, and inappropriately prescribed RRDs and had the highest Spearman coefficient. 

The BPMH is a valuable source of information for deprescription. The most important instruments for deprescription are PIM tools. The EU(7)-PIM list employed in this study presents the most comprehensive and up-to-date tool for the evaluation of PIM prescribing in Europe. It is specifically designed to cover the European drug market more appropriately than the other existing PIM criteria [7]. The prevalence of PIM use was 80% in our study which is higher than the prevalence reported in European studies in the non-hospital environment [30,31], and also higher than the prevalence determined in the study conducted in 2017 in the Clinical Hospital Dubrava [32]. A recently published study that enrolled hospitalized older patients at an internal medicine ward in Portugal detected a similar prevalence of EU(7)-PIMs (79.7%) [14]. Furthermore, analysis of this study showed that multiple medication use was the strongest predictor for PIMs, which is in line with the findings of Guillot et al. [33] and others [34,35,36]. Our results showed that recent hospitalization was also a risk factor for PIMs use. Hospitalizations increase drug use which also increases the risk of PIM prescribing. Regular evaluation of pharmacotherapy after hospitalization is necessary.

The most frequently detected EU(7)-PIM drugs were proton pump inhibitors (PPIs) (40%), previously detected in numerous studies [32,33,37]. EU(7)-PIM list consider PPI use for more than eight weeks as inappropriate for prescribing in the elderly. Long-term use of PPIs is associated with an increased risk of Clostridium difficile colitis, parietal cell hyperplasia, myopathy caused by hypomagnesemia, respiratory infections, osteoporosis-related fractures, and tubulointerstitial nephritis [7,38,39]. Elderly patients with CKD are considered to be at even higher risk of adverse effects from PPIs [11]. A study that included 2.6 million subjects outlines that PPI use was associated with a significantly increased risk of developing CKD [40]. PPIs have been highlighted as one of the three specific targets for medication optimization and deprescribing in older adults with CKD [11]. Indication for PPI use is not always clear, and its dosage and duration of use should be regularly reevaluated, especially in the elderly with renal impairment [11].

Our research found a high prevalence of potential clinically significant (C, D, X) DDIs upon admission (90.6%). Regression analysis showed that women have a higher risk for D interactions and a lower risk for X interactions as opposed to men which could be explained by the fact that drugs that depress the CNS, most often represented in D interactions, are more commonly used by women [41]. On the other hand, a lower risk for X interactions found in women can be explained by the fact that the most common interactants in X interactions were drugs indicated for chronic obstructive pulmonary disease (COPD), a condition more prevalent in men [42]. The analysis also showed a lower risk of X interactions in patients with impaired renal function, which would indicate the fact that these drugs are prescribed cautiously in this vulnerable group of patients. Cox and Snell R2 and Nagelkerke R2 measures were used to fit models in logistic regression. In terms of models, the best model was overall DDI and the weakest model was PIMs.

The most common potential consequence of the identified X interactions was an increased anticholinergic effect, with its side effects particularly high in elderly patients. Additionally, the most commonly identified clinically significant interaction between perindopril and indapamide carried an increased risk of nephrotoxicity. This result is of particular importance considering the fact that more than 40% of patients upon admission had impaired renal function (eGFR < 60 mL/min/1.73 m^2^). Recent research conducted in Croatia that included 1211 patients also found this interaction to be the most common [43]. The risk of acute renal impairment is higher when nonsteroidal anti-inflammatory drugs (NSAIDs) are added to therapy [44]. By obtaining the BPMH, a high prevalence of NSAIDs, most commonly involved in D interactions, was found. BPMH is also a key tool for OTC detection as they are not usually registered in medical documentation.

Only 11% of the study sample had normal renal function (KDIGO G1) implying the need to reconsider RRD use in the BPMH already on hospital admission. A lower level of renal function for certain drugs may require therapy adjustment and increase the risk of adverse drug events [45]. Our results showed that 64.7% of elderly patients with stages of renal impairment G3–G5 had inappropriately prescribed RRD. The prevalence was higher than the prevalence reported in an American study of elderly patients with CKD stages 3–5, but it was lower than the prevalence reported in a French study of patients aged ≥75 with eGFR < 20 mL/min/1.73 m^2^ [28,46]. 

This study compared the prescribing of inappropriately prescribed RRDs in patients with impaired renal function according to gender. Regression analysis showed that women were at higher risk of having inappropriately prescribed RRDs, which puts them at higher risk of ADEs. The epidemiology of CKD differs by gender; it reports a higher prevalence of CKD in women compared to men [47]. Faster renal function decline in men compared to women and longer life expectancy in women can partially explain the gender difference in CKD epidemiology [48]. Another risk factor contributing to the inappropriate RRD use detected in this study was the number of medications, as shown in previous studies [46,49,50].

ACEIs were the most common RRDs in this study. Inappropriate prescription of ACEIs in elderly patients with renal impairment has been noted in other studies [51,52]. ACEIs are considered superior to ARBs and other antihypertensive agents in reducing adverse renal events in non-dialyzed CKD 3–5 patients, however, the prerequisite is that they are used appropriately [53,54].

This research had certain limitations. The study included one hospital, one clinic, and was observed at only one point of care transition. Further research should also evaluate the scope of polypharmacy, PIMs, DDIs, and RRDs upon hospital discharge. The study did not include surgical patients who are considered as patients requiring more complicated therapeutic management and future research should broaden the research focus.

Despite limitations, our study suggests that the BPMH is a useful tool for detecting a wider spectrum of pharmacotherapy problems. High incidence of PIMs, DDIs, and inappropriately prescribed RRDs indicate the need for their early detection. Detection of pharmacotherapy problems upon admission is one of the crucial steps for therapy optimization during a hospital stay. The clinical pharmacist has specific pharmacotherapy knowledge and therefore, can significantly contribute to pharmacotherapy rationalization and patient safety. Although there are different decision support systems for detecting pharmacotherapy problems, they cannot adequately replace medication reconciliation and clinical pharmacists’ professional interpretation of data [55]. Decision support systems especially cannot replace a clinical pharmacist when evaluating a wider spectrum of pharmacotherapy problems, which are all positively intercorrelated and will probably occur simultaneously. We should strive for the BPMH and clinical pharmacists’ evaluation of BPMH upon admission to become the standard of health care in order to prevent the transfer and circulation of pharmacotherapy problems and to increase patient safety.

## 5. Conclusions

Clinical pharmacists’ evaluation of the BPMH showed high exposure to polypharmacy, PIMs, DDIs, and inappropriately prescribed RRDs upon hospital admission in the elderly. This study highlights the need for its detection to prevent the transfer and circulation of pharmacotherapy problems during the hospital stay, further facilitating drug optimization. The positive correlations between polypharmacy, PIMs, DDIs, and inappropriately prescribed RRDs indicate that they are not independent and that there is a greater than random probability they will occur simultaneously.

## Figures and Tables

**Table 1 pharmacy-10-00136-t001:** Patients’ characteristics.

Characteristic	Study Sample (N = 383)
Age, years, median (IQR)	76 (70–80)
Gender	
Female, n(%)	199 (52)
Body weight, kg, median (IQR)	79 (70–88)
Body height, cm, median (IQR)	168 (163–175)
Serum creatinine (µmol/L), median (IQR)	87 (68–125)
CKD-EPI (mL/min/1.73 m^2^), median (IQR)	64.4 (43.7–81.9)
eGFR stage (KDIGO classification), n(%)	
G1 Normal or high	44 (11.5)
G2 Mildly decreased	172 (44.9)
G3a Mildly to moderate decreased	68 (17.8)
G3b Moderately to severely decreased	44 (11.5)
G4 Severely decreased	30 (7.8)
G5 * Kidney failure	25 (6.5)
Residence, n (%)	
living alone	71 (18.5)
living with family/caregiver	302 (78.9)
nursing home	10 (2.6)
Admission to hospital, n(%)	343 (89.6)
emergency elective	40 (10.4)
Recent hospitalization	126 (32.9)
Mean number of diagnoses, median (IQR)	9 (6–12)
Mean number of prescription medications (BPMH), median (IQR)	8 (5–11)
Prescribed medications (BPMH), number of patients (%)	
Low medication use (<5 medications)	70 (18.3)
Polypharmacy (5–9 medications)	191 (49.9)
Excessive polypharmacy (≥10 medications)	122 (31.9)
Drug classes (ATC groups) with the most frequent therapeutic subgroups (in BPMH)	Number of medications
C Cardiovascular system	1317
C09AA ACE inhibitors	195
C07AB Beta blocking agents, selective	181
C08CA Dihydropyridine derivatives	167
C10AA HMG CoA reductase inhibitors (Statins)	157
C03CA Loop-diuretics, sulfonamides	153
A Alimentary tract and metabolism	595
A02BC Proton pump inhibitors	154
A12BA Potassium	89
A10BA Biguanides	84
N Nervous system	319
N05BA Benzodiazepines	123
B Blood and blood-forming organs	252
B01AC Platelet aggregation inhibitors	143

* There were 8 patients receiving dialysis (stage 5D) and 1 patient who underwent kidney transplantation. Abbreviations: IQR, interquartile range; CKD-EPI, Chronic Kidney Disease Epidemiology Collaboration; eGFR, estimated glomerular filtration rate; KDIGO, Kidney Disease: Improving Global Outcomes; BPMH, Best possible medication history; OTC, over-the-counter.

**Table 2 pharmacy-10-00136-t002:** The list of the most frequent EU(7)-PIMs in the BPMH.

ATC CODE	EU(7)-PIM	N = 689	%
A02BC02	pantoprazole	121	17.6
N05BA01	diazepam	75	10.9
N02AX02	tramadol	59	8.6
C02AC05	moxonidine	49	7.1
A02BA02	ranitidine	35	5.1
C02CA06	urapidil	27	3.9
C01EB15	trimetazidine	26	3.8
M01AE03	ketoprofen	24	3.5
A02BC05	esomeprazole	22	3.2
N05BA12	alprazolam	20	2.9
M01AB05	diclofenac	18	2.6
B01AF01	rivaroxaban	16	2.3
C01AA08	methyldigoxin	16	2.3
A10BH02	vildagliptin	15	2.2
C01BD01	amiodarone	12	1.7
N05BA06	lorazepam	11	1.6

Abbreviations: ATC, Anatomical Therapeutic Chemical drug classification system; PIM, potentially inappropriate medication.

**Table 3 pharmacy-10-00136-t003:** The most common potential clinically significant DDIs in the BPMH.

DDI		Number	Summary
X Interactions (with ≥ 2 Cases)			
Ipratropium	umeclidinium	3	Increased anticholinergic effects
Ipratropium	glycopyrronium	3	Increased anticholinergic effects
Ipratropium	tiotropium	2	Increased anticholinergic effects
Ipratropium	loratadine	2	Increased anticholinergic effects
Tamsulosin	urapidil	2	Synergistic pharmacotherapeutic effects (e.g., hypotension, syncope)
Doxazosin	tamsulosin	2	Alpha1-Blockers may enhance the hypotensive effect of other Alpha1-Blockers
Carvedilol	salbutamol	2	Beta-Blockers (Nonselective) may diminish the bronchodilatory effect of Beta2-Agonists
D Interactions (top 5)			
Furosemide	ibuprofen	18	NSAID may diminish the diuretic effect of Loop Diuretics. Loop Diuretics may enhance the nephrotoxic effect of NSAID
Moxonidine	bisoprolol	16	Alpha2-Agonists may enhance the AV-blocking effect of Beta-Blockers. Sinus node dysfunction may also be enhanced. Beta-Blockers may enhance the rebound hypertensive effect of Alpha2-Agonists. This effect can occur when the Alpha2 Agonist is abruptly withdrawn
Diazepam	tramadol	14	Increased risk for CNS depression
Ketoprofen	furosemide	13	NSAID may diminish the diuretic effect of Loop Diuretics. Loop Diuretics may enhance the nephrotoxic effect of NSAID
Acetylsalicylic acid	ibuprofen	12	NSAIDs (Nonselective) may enhance the adverse/toxic effect of Salicylates. An increased risk of bleeding may be associated with use of this combination. NSAIDs (Nonselective) may diminish the cardioprotective effect of Salicylates. Salicylates may decrease the serum concentration of NSAIDs (Nonselective).
C Interactions			
Indapamide	perindopril	53	Indapamide may enhance the nephrotoxicity. Indapamide may enhance hypotensive effect of ACEI

Abbreviations: DDI, drug-drug interaction, NSAIDs. nonsteroidal anti-inflammatory drugs; AV, atrioventricular; CNS, central nervous system; ACEI, Angiotensin-converting enzyme inhibitor.

**Table 4 pharmacy-10-00136-t004:** Identified RRDs in BPMH (with <3 cases are not presented).

ATC Code	Drug	RRD	Unadjusted Dose	Adjusted Dose	Contraindicated Use
C02AC05	Moxonidine	30	20	10	0
A10BA02	Metformin	29	7	16	6
C09AA05	Ramipril	17	2	15	0
B01AC06	Acetylsalicylic acid	15	0	0	15
C01EB15	Trimetazidine	15	9	3	3
C08CA13	Lercanidipine	12	3	0	9
C09BA04	Perindopril/indapamide	12	8	1	3
C09BX01	Perindopril/indapamide/amlodipine	12	0	0	12
C01AA08	Methyldigoxin	9	7	2	0
C10AA07	Rosuvastatin	9	0	5	4
C03DA04	Eplerenone	8	3	1	4
A02BA02	Ranitidine	8	3	5	0
C09BA05	Ramipril/hydrochlorothiazide	8	2	6	0
M01AE01	Ibuprofen	7	0	0	7
C09BB04	Perindopril/amlodipine	7	6	0	1
C03BA11	Indapamide	5	0	0	5
M01AE03	Ketoprofen	5	0	0	5
C09AA04	Perindopril	5	5	0	0
C09DA03	Valsartan/hydrochlorothiazide	5	0	0	5
M04AA01	Allopurinol	4	0	4	0
A10BD08	Metformin/vildagliptin	4	2	2	0
C09AA03	Lisinopril	3	1	2	0
C09BA03	Lisinopril/hydrochlorothiazide	3	2	0	1
C09BX02	Perindopril/bisoprolol	3	2	1	0
A10BH02	Vildagliptin	3	0	3	0

Abbreviations: ATC, Anatomical Therapeutic Chemical drug classification system; RRD, renal risk drug.

**Table 5 pharmacy-10-00136-t005:** Spearman correlations between criterion variables (N = 383; N_RRD_ = 167).

No	Variable	PIM	DDI	Inappropriately Prescribed RRD
1	PIM	-		
2	DDI	0.114 *	-	
3	Inappropriately prescribed RRD	0.137	0.159 *	-
4	Polypharmacy	0.372 ***	0.423 ***	0.261 ***

* *p* < 0.05; *** *p* < 0.001. Abbreviations: PIM, potentially inappropriate medication; DDI, drug-drug interaction; RRD, renal risk drug.

**Table 6 pharmacy-10-00136-t006:** The association between patients’ characteristics and excessive polypharmacy (reference: <5 Drugs), PIMs, DDIs (N = 383), inappropriately prescribed RRDs (N = 167) (Odds ratios and 95% confidence interval (CI)).

Predictor	Excessive Polypharmacy	PIM	DDI			Inappropriately Prescribed RRD
			Overall	D	X	
(Intercept)	0.05 [0–5]	0.28 [0.01–14.91]	0.77 [0–197.43]	0.29 [0.01–8.28]	1.58 [0–809.41]	0.04 [0–13.47]
Female	1.41 [0.73–2.75]	1.02 [0.57–1.83]	1.04 [0.44–2.46]	2.38 [1.49–3.8] ***	0.29 [0.12–0.73] **	2.32 [1.07–5]*
Age	1 [0.95–1.05]	1 [0.96–1.05]	0.97 [0.91–1.03]	0.98 [0.95–1.02]	0.95 [0.88–1.01]	1 [0.94–1.06]
BMI	1.02 [0.95–1.09]	1 [0.94–1.06]	1 [0.92–1.1]	0.99 [0.95–1.04]	0.95 [0.87–1.05]	0.95 [0.88–1.02]
Renal function	1.41 [1.08–1.85] *	0.87 [0.69–1.11]	1.04 [0.71–1.52]	1.02 [0.85–1.22]	0.71 [0.51–0.99] *	2.38 [1.55–3.67] ***
Elective admission type	2.52 [0.72–8.76]	0.54 [0.19–1.52]	2.12 [0.23–19.82]	0.97 [0.44–2.12]	1.44 [0.42–4.92]	0.54 [0.15–1.99]
Number of diagnoses	1.12 [1.02–1.22] *	1.01 [0.93–1.09]	1 [0.89–1.12]	1.01 [0.95–1.08]	1.07 [0.97–1.19]	1.03 [0.93–1.13]
Recent hospitalization	3.08 [1.46–6.49] **	2.4 [1.14–5.05] *	0.39 [0.14–1.05]	0.76 [0.45–1.27]	0.46 [0.18–1.17]	1.05 [0.47–2.35]
Residence±						
Living with family/caregiver	1.46 [0.62–3.45]	1.32 [0.66–2.65]	1.02 [0.38–2.74]	0.67 [0.36–1.24]	1.06 [0.27–4.09]	1.03 [0.38–2.77]
Number of prescription medications	–	1.46 [1.3–1.64] ***	2.34 [1.78–3.09] ***	1.35 [1.25–1.47] ***	1.42 [1.24–1.63] ***	1.2 [1.06–1.35] **
Cox & Snell R^2^	0.184	0.186	0.215	0.213	0.126	0.238
Nagelkerke R^2^	0.252	0.298	0.464	0.285	0.293	0.327

* *p* < 0.05; ** *p* < 0.01; *** *p* < 0.001. ±ref. Alone. Nursing home is omitted from table because the results are meaningless (eg odds ratio CI includes infinity for PIMs and overall DDIs. Abbreviations: PIM, potentially inappropriate medication; DDI, drug-drug interaction; RRD, renal risk drug.

## Data Availability

The data were used exclusively for the research conducted as part of this study and were kept confidential.

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
