# Peer review of "Pharmacotherapy Problems in Best Possible Medication History of Hospital Admission in the Elderly"

_pharmacy, 2022, doi:10.3390/pharmacy10050136_

Round 1

Reviewer 1 Report

Thank you for the very well written and understandable article. Overall, I have only a few comments of an editorial and content-related nature.

Line 17: Remove spaces between numbers and percentages.

Lines 18 and 22: Write out the abbreviations eGFR and BPMH in the abstract, as they are only mentioned once.

Line 101: I have not read anything before about the recording of the diagnoses of the study participants. Were these also recorded as part of the BPHM? And were they filtered for relevant (chronic) and rather irrelevant diagnoses? This is very important for the interpretation of the regression. For me, rather unexpectedly, the number of diseases in the regression does not have a very large influence.

Table 1: Please correct "diagnosis" to "diagnoses" in the text, as the plural is always meant.

Line 200: Please correct "rampiril" to "ramipril".

Tables 3 and 6: Please check the formatting of the tables again. Tables 3 and 6 contain line breaks that make reading difficult (e.g. in the confidence intervals in table 6). If necessary, set a smaller font size for the tables. This is usually sufficient.

Line 230: Please check the language again throughout the discussion. For example, line 263: A recently published study...

Line 284: Please discuss the model fit of the regression in the discussion.

Line 299: 1,299 instead of 1299. Please use the correct decimal separators throughout the text and in tables (especially Tab. 6). Example: 1,477,386.12[0-Inf]

Line 327: In some sources it is pointed out that for the estimated glomerular filtration rate older people (>75 years) were not sufficiently considered in validation studies and therefore the eGFR is not a reliable parameter. Please briefly discuss this in the limitations or name a source that contradicts this limitation. Or how you have dealt with this limitation.

Line 372: Two lines with reference 1. Please correct.

Author Response

We thank the reviewers for finding our work of interest and for their valuable comments and suggestions. We have incorporated the suggestions into this revised version of the manuscript. We provide a detailed list of our responses here (not bolded) to reviewers' comments (bolded):

Reviewer 1

Thank you for the very well written and understandable article. Overall, I have only a few comments of an editorial and content-related nature.

Line 17: Remove spaces between numbers and percentages.

According to Reviewer’s suggestion, spaces between numbers and percentages were removed.

Lines 18 and 22: Write out the abbreviations eGFR and BPMH in the abstract, as they are only mentioned once.

The abbreviation BPMH was described in line 12, and then the abbreviation BPMH was again mentioned in line 22. As estimated glomerular filtration rate was mentioned only once in the abstract (line 18), eGFR abbreviation was deleted.

Line 101: I have not read anything before about the recording of the diagnoses of the study participants. Were these also recorded as part of the BPHM? And were they filtered for relevant (chronic) and rather irrelevant diagnoses? This is very important for the interpretation of the regression. For me, rather unexpectedly, the number of diseases in the regression does not have a very large influence.

Diagnoses were recorded as part of BPMH. All determined diagnoses were taken into account. Patients’ diagnoses were classified according to the International Classification of Diseases (ICD-10). The data set did not contain information on the relevance of diagnoses. Determination of diagnoses is added in section Method 2.2. Data collection: „Patients’ diagnoses were classified according to the International Classification of Diseases (ICD-10).“

Table 1: Please correct "diagnosis" to "diagnoses" in the text, as the plural is always meant.

According to Reviewer’s suggestion, „diagnosis“ was corrected to „diagnoses“ in the text.

Line 200: Please correct "rampiril" to "ramipril".

According to Reviewer’s suggestion, „rampiril“ was corrected to „ramipril“.

Tables 3 and 6: Please check the formatting of the tables again. Tables 3 and 6 contain line breaks that make reading difficult (e.g. in the confidence intervals in table 6). If necessary, set a smaller font size for the tables. This is usually sufficient.

According to Reviewer’s suggestion, smaller font size was set for the tables.

Line 230: Please check the language again throughout the discussion. For example, line 263: A recently published study...

The language was checked throughout the discussion. The sentence „Recently published study which enrolled hospitalized older patients at internal medicine ward in Portugal detected similar prevalence of EU(7)-PIMs (79.7%)“ was amended to „A recently published study which enrolled hospitalized older patients at internal medicine ward in Portugal detected similar prevalence of EU(7)-PIMs (79.7%)“.

Line 284: Please discuss the model fit of the regression in the discussion.

According to Reviewer’s suggestion, the model fit of the regression is discussed in section Discussion: „Cox & Snell R2 and Nagelkerke R2 measures were used to fit models in logistic regression. In terms of models, the best model was overall DDI and the weakest model was PIMs.“

Line 299: 1,299 instead of 1299. Please use the correct decimal separators throughout the text and in tables (especially Tab. 6). Example: 1,477,386.12[0-Inf]

According to Reviewer’s suggestion, the correct decimal separator was used throughout the text, eg 1,211.

Line 327: In some sources it is pointed out that for the estimated glomerular filtration rate older people (>75 years) were not sufficiently considered in validation studies and therefore the eGFR is not a reliable parameter. Please briefly discuss this in the limitations or name a source that contradicts this limitation. Or how you have dealt with this limitation.

Official guidelines do not distinguish the application of equation for renal impairment according to age groups (˂65 years, 65-75 years, ˃75years).

Kidney Disease: Improving Global Outcomes (KDIGO) guidelines state: „Most equations used to estimate GFR have been primarily developed in younger populations, although subgroup analyses show that these equations perform reasonably well in older people“.

According to recommendations of Croatian society of medical biochemistry and laboratory medicine laboratories must implement the 2009 CKD-EPI equation for eGFR into routine laboratory work. The recommendations state: „Although the eGFR value is < 60 mL/min/1.73 m2 very common in the elderly (5), for prediction increased risk of adverse clinical outcomes in the use of an age-dependent diagnostic value is not recommended for adults.“

Line 372: Two lines with reference 1. Please correct.

According to Reviewer’s suggestion, one line with reference 1 was deleted.

Reviewer 2 Report

The authors attempted to elucidate the pharmacotherapy problems in elderly patients.

Polypharmacy is a critical problem, especially in patients with older patients. It is sometimes difficult to solve this problem because of lacking a good strategy.

In my opinion, this study is worthwhile to tackle the problem.

However, there are several concerns in this study.

First, kidney function is very important to consider polypharmacy, and the authors showed the percentage of patients by eGFR stages. The authors showed that 25 patients were categorized as G5, but no patients were categorized as CKD 5D (patients need renal replacement therapy).

If there were patients on dialysis, please show the result. Additionally, the number of patients who underwent kidney transplantation should be shown.

Second, I cannot understand how the authors selected to show the categories of drugs as the most common drug class in Table 1. Were these dependent on the number of prescribed cases? In my opinion, the bottom “ Respiratory system 182” does not make sense to show. I think that the authors should reconsider showing the drug categories in Table 1.

Third, Table 6 is difficult to read. The odds ratios for “Nursing home” might be meaningless due to the wide ranges of 95% CI. If the authors do not regard “Nursing home” as an important parameter, it would be better to show the results in the footnote.

Fourth, I want to know some examples of how the authors acted against the pharmacotherapy problems if they found some drug-drug interactions and contraindications.  

Author Response

We thank the reviewers for finding our work of interest and for their valuable comments and suggestions. We have incorporated the suggestions into this revised version of the manuscript. We provide a detailed list of our responses here (not bolded) to reviewers' comments (bolded):

Reviewer 2

The authors attempted to elucidate the pharmacotherapy problems in elderly patients.

Polypharmacy is a critical problem, especially in patients with older patients. It is sometimes difficult to solve this problem because of lacking a good strategy.

In my opinion, this study is worthwhile to tackle the problem.

However, there are several concerns in this study.

First, kidney function is very important to consider polypharmacy, and the authors showed the percentage of patients by eGFR stages. The authors showed that 25 patients were categorized as G5, but no patients were categorized as CKD 5D (patients need renal replacement therapy).

If there were patients on dialysis, please show the result. Additionally, the number of patients who underwent kidney transplantation should be shown.

According to Reviewer’s suggestion, the number of patients on dialysis and the number of patients who underwent kidney transplantation is added in footnote of the Table 1. “There were 8 patients receiving dialysis (stage 5D) and 1 patient who underwent kidney transplantation.”

Second, I cannot understand how the authors selected to show the categories of drugs as the most common drug class in Table 1. Were these dependent on the number of prescribed cases? In my opinion, the bottom “ Respiratory system 182” does not make sense to show. I think that the authors should reconsider showing the drug categories in Table 1.

The section „The most common drug classes (ATC groups) with the most frequent therapeutic subgroups (in BPMH)“ in Table 1 presents the ten most common therapeutic subgroups grouped into ATC groups with regard to the number of reported cases in BPMH. At the same time, the presented ATC groups correspond to the most common ATC groups. According to the reviewer's suggestion, the ATC group Respiratory system was removed from the table. We believe it is important to show the patterns of medication use and frequency.

The sentence in Results „According to the ATC classification, the most frequent drug classes in BPMH were groups C (cardiovascular system), A (alimentary tract and metabolism), N (nervous system), B (blood and blood-forming organs) and R (respiratory system.“ was amended to „According to the ATC classification, the most frequent drug classes in BPMH were groups C (cardiovascular system), A (alimentary tract and metabolism), N (nervous system) and B (blood and blood-forming organs).“

Third, Table 6 is difficult to read. The odds ratios for “Nursing home” might be meaningless due to the wide ranges of 95% CI. If the authors do not regard “Nursing home” as an important parameter, it would be better to show the results in the footnote. 

According to Reviewer’s suggestion, the „Nursing home“ was mentioned in the footnote f the Table 6. as follows: „Nursing home is omitted from table because the results are meaningless (eg odds ratio CI includes infinity for PIMs and overall DDIs)“. The following sentence has been added to the Results section (3.7. Factors associated with polypharmacy, PIMs, DDIs and inappropriately prescribed RRDs): „It should be noted that the residence in nursing home has an extremely high and insignificant odds ratio, which is due to the low number of participants who have no variability in the criterion variable.“           

Fourth, I want to know some examples of how the authors acted against the pharmacotherapy problems if they found some drug-drug interactions and contraindications.  

Pharmacist interventions regarding drug-drug interactions and contraindications were referred to the physician. The analysis of pharmaceutical intervention may be the subject of the next research. In this paper we decided to focus on the extent of pharmacotherapy problems that patients have when entering the hospital system.